# Abnormal Head Size in Children and Adolescents with Congenital Nervous System Disorders or Neurological Syndromes with One or More Neurodysfunction Visible since Infancy

**DOI:** 10.3390/jcm9113739

**Published:** 2020-11-20

**Authors:** Lidia Perenc, Agnieszka Guzik, Justyna Podgórska-Bednarz, Mariusz Drużbicki

**Affiliations:** 1Department of Physiotherapy, Institute of Health Sciences, Medical College, University of Rzeszów, 35-959 Rzeszów, Poland; lidiaiadam.perenc@wp.pl (L.P.); j.e.podgorska@gmail.com (J.P.-B.); mdruzb@ur.edu.pl (M.D.); 2Centre for Innovative Research in Medical and Natural Sciences, University of Rzeszow, Warzywna 1a, 35-310 Rzeszow, Poland

**Keywords:** microcephaly, macrocephaly, neurodysfunction, epilepsy

## Abstract

The current study was designed to investigate co-occurrence of absolute/relative microcephaly, absolute/relative macrocephaly and congenital nervous system disorders or neurological syndromes with symptoms visible since infancy, based on fundamental data acquired during the admission procedure at a neurological rehabilitation ward for children and adolescents. The study applied a retrospective analysis of data collected during the hospitalization of 327 children and adolescents, aged 4–18 years, affected since infancy by congenital disorders of the nervous system and/or neurological syndromes associated with a minimum of one neurodysfunction. To identify subjects with absolute/relative microcephaly, absolute/relative macrocephaly in the group of children and adolescents, the adopted criteria took into account z-score values for head circumference (z-score hc) and head circumference index (z-score HCI). Dysmorphological (x+/−3s) and traditional (x+/−2s) criteria were adopted to diagnose developmental disorders of head size. Regardless of the adopted criteria, absolute macrocephaly often coexists with state after surgery of lumbar myelomeningocele and hydrocephalus, isolated hydrocephalus, hereditary motor and sensory polyneuropathy, and Becker’s muscular dystrophy (*p* < 0.001, *p* = 0.002). Absolute macrocephaly is often associated with neural tube defects and neuromuscular disorders (*p* = 0.001, *p* = 0.001). Relative microcephaly often occurs with non-progressive encephalopathy (*p* = 0.017, *p* = 0.029). Absolute microcephaly, diagnosed on the basis of traditional criteria, is often associated with epilepsy (*p* = 0.043). In children and adolescents with congenital nervous system disorders or neurological syndromes with one or more neurodysfunction visible since infancy, there is variation in abnormal head size (statistically significant relationships and clinical implications were established). The definitions used allowed for the differentiation of abnormal head size.

## 1. Introduction

Microcephaly and macrocephaly are developmental disorders of the head (DDH) and are also treated as clinical symptoms. In order to obtain a diagnosis, the size of the head is estimated by measuring head circumference (hc), namely the occipito-frontal circumference (ofc), and comparing it to a biological reference frame [1,2]. Next, it is necessary to determine whether the hc is proportionally aberrant, i.e., microcephaly, or macrocephaly. There are two criteria differentiating microcephaly and macrocephaly—used in dysmorphology (hc < x − 3s, hc > x + 3s, respectively) and traditional methodology (hc < x − 2s, hc > x + 2s, respectively), often used in clinical practice [1]. The use of hc alone is considered insufficient for diagnosis [1,3,4].

The head circumference index (HCI) and Michalski’s classification can be used to assess the size of the head^4^. For example, children and adolescents from the city of Rzeszow (Poland) have differentiated head sizes in different age groups. The sizes of heads in boys and girls of different age groups were classified with consideration of parameters such as hc and body height (h), the correlation between them, and Michalski’s classification. In boys aged 4–11, head size was assessed as very large, in boys aged 12–13—large, in boys aged 14—medium, and in boys aged 15–18—small. In girls aged 4–11, the head size was described as very large, in girls aged 12—large, in girls aged 13–17—medium, and in girls aged 18—small [3]. From a clinical point of view, this type of head size classification reflects the process of differentiating body proportions in children and adolescents, but still remains unsatisfactory. Some authors, apart from the relative microcephaly/macrocephaly, characterized by a proportional decrease/increase in hc, h and body weight, distinguish absolute microcephaly/macrocephaly, characterized by an isolated decrease/increase in hc [1]. It is also important that hc and h are measurements determined by the size of the skeleton, while body weight does not only reflect the size of the skeleton, and is determined to a lesser extent genetically than h [5].

The terms presented above are not clearly defined. This article proposes definitions of relative and absolute microcephaly, as well as relative and absolute macrocephaly, based on hc, HCI, and adopts dysmorphological and traditional criteria [1]. The definitions presented in this article are completely different than the ones traditionally used. It was deliberated over which of them to use in daily practice.

Abnormal head size (AHS) occurs in diseases with damage to the nervous system [1]. Disorders of the nervous system can lead to several neurodysfunctions; in such cases, it is possible (but not certain) for encephalopathy to occur. Encephalopathy, largely defined as any damage of the brain, can arise from various factors. Crucially, it can affect the start of school education, for it causes disorders of behaviour, cognitive functions, and motor abilities [6,7,8].

AHS is a worrying symptom that can be determined on the basis of physical examination [1,8]. Diagnosing diseases with damage to the nervous system in children and adolescents is not an easy task [9,10,11]. Appropriate clinical deduction and an appropriately planned diagnostic and differential process can assure success and proper diagnosis [12]. Our observations show that a significant number of children and adolescents with congenital nervous system disorders or neurological syndromes with one or more neurodysfunctions visible since infancy have escaped diagnosis (Figure 1). 

The aim of the study is to determine whether the presented definitions will allow differentiation of AHS in children and adolescents with congenital disorders of the nervous system or neurological syndromes with one or more neurological disorders visible from infancy. Moreover, the aim of the study is seeking the relationship between relative and absolute microcephaly, relative and absolute macrocephaly and units or disease syndromes with neurodysfunction in a group of children, based on data collected during the admission procedure to the Department of Children and Youth Neurological Rehabilitation and establishing clinical implications to facilitate the diagnostic-differential process. Additionally, this study aims to identify which of the traditional or dysmorphological criteria differentiates AHS better.

## 2. Materials and Methods 

### 2.1. Participants

The retrospective study took into account information related to 327 children and adolescents admitted between 2012 and 2016 to the Neurological Rehabilitation Ward for Children and Adolescents in Regional Hospital No. 2 in Rzeszow, Poland, and staying at the Clinical Regional Rehabilitation and Education Centre. All the patients eligible for the study were hospitalized during the period from 2012 to 2016 and presented congenital nervous system disorders or neurological syndromes with one or more neurodysfunctions visible since infancy. The following additional eligibility criteria were adopted: ages between 4–18 years (4 years of age was the beginning of the analyzed age range for three reasons: (a) because of the high percentage of children who are under 4 years of age without diagnosis [9,10,11], (b) because of the methodology adopted in the entire research project [2,13], (c) due to the presence of a biological frame of reference for the examined age range containing statistical characteristics of h [14], hc [3], HCI [13]), informed consent from both the children and their parents/legal guardians, availability of the measurements of h, and hc, as well as complete diagnostic data (recognition, rating on scale: Gross Motor Function Classification System—GMFCS), all of which were acquired during a single admission procedure.

Patients were excluded from the study if they had no diagnosis of a congenital disorder of the nervous system or a neurological syndrome linked with one or more neurodysfunction visible from infancy, or if they presented combinations of congenital disorders of the nervous system or the neurological syndromes (e.g., Down syndrome, neural tube defect, or phenylketonuria co-occurring with cerebral palsy). Additionally, patients were excluded if they were not hospitalized in the relevant period, or if there was more than one admission procedure, if they fell into the age groups of below 4 or above 18 years old (due to the lack of biological frame of reference containing statistical characteristics of h, hc, HCI), if their records did not contain complete diagnostic information and/or anthropometric measurements (h, hc), and finally, if no informed consent was given by the children and their parents/legal guardians. Such a selection of subjects, inclusion and exclusion criteria were accepted earlier [13]. 

From 2012 to 2016, a total of 2637 hospitalizations took place in the Neurological Rehabilitation Ward for Children and Adolescents at the Regional Hospital No. 2 in Rzeszow, Poland, and at the Clinical Regional Rehabilitation and Education Centre (KRORE). Of these, 327 patients were found to meet the inclusion criteria. As a result, the retrospective analysis took into account 327 children (143 girls—43.7%, 184 boys—56.3%), with a mean age of 9.7 ± 4.3 years (median 9.0 years; the youngest child was 4 and the oldest was 18 years of age).

The study protocol was accepted by the Bioethics Commission at the University of Rzeszow, Poland, and the procedures used complied with the applicable guidelines and regulations. Before the application was filed with the bioethics committee, informed consents were obtained from the patients and their parents/legal guardians as well as the director of the hospital.

### 2.2. Procedures and Data Analyses

The basic data taken into account in the retrospective analysis included the patients’ age, sex, as well as diagnosis, GMFCS, h, and hc. All the information was retrieved from the patient records collected at admission.

The diagnoses had been specified by neurologists, geneticists, endocrinologists, and other specialists before admission to KRORE. According to the medical records, the children presented a variety of conditions or syndromes associated with damage to the nervous system. All of these were congenital anomalies and/or disorders, with or without encephalopathy, and accompanied with motor defects (neurodysfunctions) visible from early childhood. The criteria reported in the related literature (i.e., suspected encephalopathy or no encephalopathy, its etiopathogenesis and nature)^8^ where applied in dividing the patients into subgroups (Table 1A). Seven subgroups were separated from the entire study group: six with diseases/syndromes usually involving encephalopathy: progressive metabolic diseases (MD), progressive epilepsy—genetically conditioned epileptic syndromes (EE), non-progressive in neural tube defects (NTDs), non-progressive in genetically conditioned diseases (chromosomal aberrations, monogenic diseases, except for neuromuscular diseases) (GD), non-progressive toxic (TE), non-progressive in cerebral palsy (CP) and one in the diseases usually without encephalopathy: neuromuscular diseases (NMD)—given the nature and presence of expected encephalopathy [8]. Smaller subgroups were merged into two larger ones: with progressive encephalopathy (PE), non-progressive (NPE), and neuromuscular diseases (NMD). Due to the large variety of neural tube defects [15] and significance of further surgical treatment [16,17,18], this subgroup was divided into those operated on for myelomeningocele and hydrocephalus (sasMMC and HCP), operated on only because of myelomeningocele (sasMMC), and other cases where no surgical treatment was given. The subgroup with GD included both chromosomal aberrations and genetic mutations. It should be mentioned that some authors point to chromosomal disorders and genetic mutations as causes of short stature [19,20]. Down syndrome and Prader-Willi syndrome also belong to this group, alongside with other genetically determined diseases, including mutations of a single gene [19,20].

Analysis of the diagnoses showed the types of CP, as proposed by Hagber [8]. Principal diagnoses were accompanied with additional diagnoses: symptomatic epilepsy and hypothyroidism. All the children with epilepsy and hypothyroidism were taking medication [13]. 

The severity of disability in the entire study group is based on the five-step Gross Motor Function Classification System—GMFCS I-V [21,22]. In order to be statistically significant, I and II levels of GMFCS were joined as group A, IV and V levels–as group C, and level III corresponded to group B—GMFCS A-C [23] (Table 1B). In Poland, each patient with neurodysfunction (with CP and without CP) admitted to the Neurological Rehabilitation Ward for Children and Adolescents is assessed on the GMFCS scale [24,25].

The relevant anthropometric measurements had been carried out by the hospital personnel in compliance with the guidelines approved at KRORE. HCI, i.e., a quotient of head circumference and body height (hc/h), were calculated for each patient. In order to assess developmental deficits based on all of the previously mentioned parameters, z-scores were calculated for h (z-score h), hc (z-score hc) and HCI (z-score HCI). Normative values published earlier were applied as a reference frame [3,14] (Table 2A). The z-score indicators were used to identify the developmental disorders described below. A similar use of the z-score indicators had been approved previously [13,23,26].

Using two assessment criteria in the study group, i.e., dysmorphological and traditional ones used in clinical practice, distinguished children and adolescents with the correct head size, microcephaly, and macrocephaly [1] (Table 2B). In addition, a modification has been proposed. Next to hc, HCI was used. In this way, the differences in body proportions in terms of hc and h were taken into account. The rules adopted for recognizing AHS are presented in Table 2C. In order to make an assessment based on all of the above criteria, for each subject was calculated a head circumference z-score (z-score hc), and a head z-score ratio of the head circumference and body height (z-score HCI). Previously published normative values were used as the reference frame^3^.

Relationships between the coexistence of DDH were sought, i.e., relative microcephaly, absolute microcephaly, relative macrocephaly, absolute macrocephaly, disease entities and syndromes with neurodysfunction and with separate subgroups, as well as within separate subgroups. Additional diagnoses (symptomatic epilepsy as opposed to genetically determined epileptic syndrome, hypothyroidism) and GMFCS score were also considered.

The dependence analysis was presented in the form of a summary of the number (N) and percentage structure (*N*%) of answers to selected questions in the compared groups. In the crosstabs, adjusted standardized residuals (ASR) are presented next to percentages. Values higher than 1.96 correspond to a greater number, and those lower than −1.96 represent a smaller number than random distribution. Statistical inference methods were used to determine in what way the intergroup differences reflect certain regularities in the relevant population, or whether they are random. Due to the nominal nature of the characteristics being compared, a chi-square test of independence was further applied. Nominal regression was used to assess the relationships between the dependent qualitative and independent quantitative variables. In order to determine correlation between two variables that did not meet the criterion of normal distribution, we used the Spearman’s rank correlation coefficient. The value of *p* < 0.05 was assumed to reflect statistical significance. Pearson’s Contingency Coefficient C (Cp) can only take positive values (Cp ≥ 0). A relationship is indicated by Cp far from 0, while values approaching 1 show a perfect association. 

## 3. Results

The percentages for the seven separated subgroups are presented as follows: MD 2.1%, EE 0.3%, NTDs 7.3%, GD 7.0%, TE 0.3%, CP 73.1% and NMD 9.8%. In division into 3 previously defined subgroups are as follows: PE 2.4%, NPE 88.1%, and NMD 9.8% [13].

Characteristics regarding diagnoses, subgroups, number of subjects *(N*), their percentage (*N*%) and abbreviations used are listed in Table 1A.

In the study group, those with CP constituted the largest group (73.1%). Analysis of the diagnoses showed the following types of CP, as proposed by Hagber [8,13]: spastic—93.7% (*N* = 223), mixed 5% *(N* = 12), ataxic 1.7% (*N* = 4). No cases of dyskinetic type were identified. Among those with spastic CP, 34.1% (*N* = 76) presented with tetraplegia, 40.4% with diplegia (*N* = 90), and 25.6% with hemiplegia (*N* = 57). Principal diagnoses were accompanied with additional diagnoses: symptomatic epilepsy 26.3% (*N* = 86), and hypothyroidism 4.3% (*N* = 14).

The statistical characteristics of the GMFCS score in the entire study group are presented in the Table 1C and in the subgroup with CP—in the Table 1D. The presented results are similar in both cases.

Selected quantitative characteristics of indicators are presented: z-score hc, z-score h, z-score HCI; and such as arithmetic mean ($\bar{x}$), median (Me), standard deviation (s), smallest (Min) and largest value (Max), 25th (c25) and 75th centile (c75) (Table 2A). 

The mean and median values for all z-scores are generally lower than zero, except for the z-score HCI. In the examined group, compared to the reference system, the values of anthropometric features characterizing the growth process: hc, h were lower. The distribution of hc and HCI in this specific population is shown in Figure 2A,B.

Given the dysmorphological criteria, 83.5% of patients had normal head size and 16.5% had abnormal one. Among the subjects with AHS, the most common were relative microcephaly—66.7%, rarely absolute macrocephaly—14.8%, and absolute microcephaly and relative macrocephaly—9.3% each (Table 2D). Considering traditional criteria, normal head size was present in 68.5% of respondents and AHS in 31.5% respectively. Among the subjects with an incorrect head size, relative microcephaly was similar—58.3%, as before less frequently absolute macrocephaly—16.5%, relative macrocephaly—13.6%, absolute microcephaly—11.7% (Table 2E).

The following relationship was investigated: the size of the head—dysmorphological and traditional classification (hc) and z-score HCI (Table 2F) in the study group. In the case of the adopted dysmorphological criterion of head size (hc), it was found that the higher the z-score hc/h values, the lower the chance of microcephaly (*p* = 0.001), and the higher the occurrence of macrocephaly (*p* = 0.004). As the mean HCI z-score increases, the risk of microcephaly decreases (OR = 0.743), and the risk of macrocephaly increases (OR = 1.446). In the case of the adopted criterion of traditional head size (hc), it was found that the higher the mean values of the HCI z-score, the lower the chance of microcephaly (*p* = 0.028), and the greater the occurrence of macrocephaly (*p* = 0.006). As the mean HCI z-score increased, the risk of microcephaly decreased (OR = 0.854), and the risk of macrocephaly increased (OR = 1.291) (Table 2F).

Relationships between the coexistence of DDH with: relative microcephaly, absolute microcephaly, relative macrocephaly, absolute macrocephaly and with main recognition, and with separate subgroups, as well as within separate subgroups and with additional diagnoses, are presented below.

The presence of more/less frequent co-occurrence and the presence of statistically significant relationships was found between:AHS—dysmorphological classification (hc and HCI) and units and syndromes running with neurodysfunction (*p* < 0.001, Cp = 0.736). Relative microcephaly often coexisted with CP (*N*% = 75.6%, ASR = 2.5), and rarely with sasMMC and HCP (*N*% = 0.0%, ASR = −2.0). Relative macrocephaly coexisted frequently with ACM (*N*% = 100.0%, ASR = 3.2), with PMS (*N*% = 100.0%, ASR = 3.2), and rarely with CP (*N*% = 4.9%, ASR = −2.0). Absolute macrocephaly often coexisted with sasMMC and HCP (*N*% = 100.0%, ASR = 3.5), with HCP (*N*% = 100.0%, ASR = 2.4), HMSN (*N*% = 100%, ASR = 2.4) with BMD (*N*% = 100.0%, ASR = 2.4), and rarely with CP (*N*% = 7.3%, ASR = −2.8) (Table 3A).

2.AHS—traditional classification (hc and HCI) and units and syndromes running with neurodysfunction (*p* = 0.002, Cp = 0.679). Relative microcephaly coexisted frequently with DS (*N*% = 100.0%, ASR = 2.1), and rarely with sasMMC and HCP (*N*% = 0.0%, ASR = −2.1), in children with LGMD (*N*% = 0.0%, ASR = −2.7). Absolute microcephaly often coexisted with CP (*N*% = 16.7%, ASR = 2.4). Relative macrocephaly coexisted frequently with LGMD (*N*% = 80.0%, ASR = 4.4), and with PMS (*N*% = 100.0%, ASR = 2.5). Absolute macrocephaly coexisted often in children with sasMMC and HCP (*N*% = 100.0%, ASR = 4.0), z HCP (*N*% = 100.0%, ASR = 2.3), sasMMC (*N*% = 100.0%, ASR = 3.2), z HMSN, BMD, DMD (*N*% = 100.0%, ASR = 2.3), and rarely with CP (*N*% = 9.7%, ASR = −2.8) (Table 3B). Cp was higher for dysmorphological classification (0.736) than the traditional method (0.679) (Table 3A,B). In this case, the dysmorphological classification better differentiated the relationship between abnormal head size and individual diseases/syndromes.3.AHS—dysmorphological classification (hc and HCI) and subgroups based on the classification with regard to etiopathogenesis, presence and character of encephalopathy (*p* = 0.001, Cp = 0.590). Relative macrocephaly coexisted often with CP (*N*% = 75.6%, ASR = 2.5). Absolute microcephaly coexisted rarely with NTDs (*N*% = 0.0%, ASR = −2.9). It also rarely coexisted in the subgroup NMD (*N*% = 0.0%, ASR = −2.5). Relative macrocephaly coexisted less frequently with CP (*N*% = 4.9%, ASR = −2.0). Absolute macrocephaly frequently coexisted in subgroup with NTDs (*N*% = 75.0%, ASR = 3.5) and NMD (*N*% = 66.7%, ASR = 2.6), less frequently in subgroup with CP (*N*% = 7.3%, ASR = −2.8) (Table 3C).4.AHS—traditional classification (hc and HCI) and subgroups based on classification with regard to etiopathogenesis, presence and character of encephalopathy (*p* = 0.001, Cp = 0.525). Relative microcephaly more frequently coexisted in subgroup with GD (*N*% = 90.0%, ASR = 2.1), coexisted rarely in subgroup with NTDs (*N*% = 22.2%, ASR = −2.3) and in subgroup with NMD (*N*% = 20.0%, ASR = −2.6). Absolute microcephaly frequently coexisted in group with CP, (*N*% = 16.7%, ASR = 2.4). Relative macrocephaly coexisted frequently in children with NTDs (*N*% = 66.7%, ASR = 4.2) and with NMD (*N*% = 40.0%, ASR = 2.1), it was rare in group with CP (*N*% = 9.7%, ASR = −2.8) (Table 4A). Cp was higher for the newer dysmorphological classification (0.590) than the traditional method (0.525) (Table 3C and Table 4A). In this case, the dysmorphological classification better differentiates the relationship between abnormal head size and seven subgroups based on the classification taking into account the etiopathogenesis, presence and nature of encephalopathy.

5.AHS—dysmorphological classification (hc and HCI) and subgroups based on classification with regard to presence and character of encephalopathy (*p* = 0.017, Cp = 0.398). Relative microcephaly was common in the NPE subgroup (*N*% = 70.6%, ASR = 2.5), and rare in the NMD subgroup (*N*% = 0.0%, ASR = −2.5). The relationship is statistically significant. Absolute macrocephaly was frequent in the NMD subgroup (*N*% = 66.7%, ASR = 2.6), and rare in the NPE subgroup (*N*% = 11.8%, ASR = −2.6) (Table 4B).6.AHS—traditional classification (hc and HCI) and subgroups based on classification with regard to presence and character of encephalopathy (*p* = 0.029, Cp=0.347). Relative microcephaly coexisted rarely in the subgroup with NMD (*N*% = 20.0%, ASR = −2.6). Relative macrocephaly coexisted frequently in the subgroup with NMD (*N*% = 40.0%, ASR = 2.6), and rarely in subgroup with NPE (*N*% = 10.9%, ASR = −2.3). Absolute macrocephaly frequently occurs in the subgroup with NMD (*N*% = 40.0%, ASR = 2.1) (Table 4C). Cp value is higher for dysmorphological classification (0.398) than for traditional one (0.347) (Table 4B,C). In this case, the criterion of +/− 3s better differentiated the relationship between abnormal head size and three subgroups distinguished based on the classification taking into account the presence and character of encephalopathy.7.AHS—dysmorphological classification (hc and HCI) and types of CP (*p* = 0.029, Cp = 0.425. Absolute microcephaly often coexisted with a mixed form of CP (*N*% = 66.7, ASR = 3.0), and rarely with its spastic type (*N*% = 5.3%, ASR = −3.0). Absolute macrocephaly often occurred with the spastic form of CP (*N*% = 7.9, ASR = 3.5), and rarely with a mixed form (*N*% = 0.0%, ASR = −3.5) (Table 4D).8.AHS—traditional classification (hc and HCI) and types of CP (*p* = 0.016, Cp = 0.311). Absolute microcephaly often occurred with a mixed form of CP (*N*% = 60.0%, ASR = 2.7), and rarely with the spastic form (*N*% = 13.6%, ASR = −2.3). Relative macrocephaly frequently occurred with the atactic form of CP (*N*% = 100.0%, ASR = 2.8) (Table 4E). Pearson’s contingency coefficient was higher for the definition based on the three standard deviation criteria and amounts to 0.425, than for the definition based on the two standard deviation criteria (Cp = 0.311). The definition based on the criteria of three standard deviations better differentiated abnormal head size among the different forms of CP (Table 4D,E). The +/− 3s criterion in this case differentiates better the relationship between head size and cerebral palsy types.9.AHS—traditional classification (hc and HCI) and epilepsy (*p* = 0.043, Cp = 0.271). Absolute microcephaly rarely coexisted with the absence of epilepsy (*N*% = 6.5%, ASR = −2.0), and frequently with epilepsy (*N*% = 19.5%, ASR = 2.0). Absolute macrocephaly frequently coexisted with the absence of epilepsy (*N*% = 22.6%, ASR = 2.0), and rarely with epilepsy itself (*N*% = 7.5%, ASR = −2.0) (Table 5A).

The presence of more/less frequent co-occurrence and an absence of statistically significant relationships was found between:AHS—traditional classification (hc and HCI) and kind of spastic type (*p* = 0.312). Relative microcephaly frequently occurred with tetraplegia (*N*% = 78.6%, ASR = 2.0)—Table 5B.AHS—traditional classification (hc and HCI) and hypothyroidism (*p* = 0.207). Relative microcephaly and hyperthyreosis frequently coexisted (*N*% = 88.9%, ASR = 2.0). Relative microcephaly and the lack of hyperthyreosis coexisted rarely (*N*% = 55.3%, ASR = −2.0)—Table 5C.

No more/less frequent co-occurrence and no statistically significant relationships were found between:AHS—dysmorphological classification (hc and HCI) and form of spastic type (Table 5D).AHS—dysmorphological classification (hc and HCI) and epilepsy (Table 5E).AHS—dysmorphological classification (hc and HCI) and hypothyroidism (Table 5F).

A statistically significant relationship was obtained (*p* = 0.000–0.039): the higher the hc, HCI values (the larger the head size), the lower the GMFCS I–V/A–C values (the lower the level of disability) (Table 6A). Relationships were sought between the coexistence of DDH with relative microcephaly, absolute microcephaly, relative macrocephaly, absolute macrocephaly and with level of GMFCS. The presence of less frequent co-occurrence and the presence of statistically significant relationships were found between: AHS—traditional classification (hc and HCI) and level of GMFCS A–C (*p* = 0.039, Cp = 0.34). Relative microcephaly rarely coexisted with GMFCS A (*N*% = 48.3%, ASR = −2.3), and relative macrocephaly rarely coexisted with GMFCS C (*N*% = 2.6%, ASR = −2.5) (Table 6E). In other cases, there was no statistically significant relationship (Table 6B–D,F–I).

## 4. Discussion

Depending on the adopted dysmorphological/traditional criteria, the incidence of AHS was 16.5/31.5%, respectively, among children and adolescents with congenital nervous system disorders or neurological syndromes with one or more neurodysfunction visible since infancy. In more than half of the cases, AHS was represented by relative microcephaly. Microcephaly is a common symptom in clinical practice [1]. The prevalence of microcephaly (hc < x − 3s) in the general population has been estimated at approximately 1/1000 [27], and that of fetal microcephaly is 1–6/10,000 [28]. The incidence of relative microcephaly in the study group, depending on the adopted dysmorphological/traditional criterion, was respectively 11.0/18.3%. Absolute microcephaly was much less common. The distinction between relative and absolute microcephaly established that the latter, defined on the basis of traditional criteria, is statistically significantly more likely to coexist with symptomatic epilepsy among children and adolescents with congenital nervous system disorders or neurological syndromes with one or more neurodysfunction visible since infancy. Other authors indicate the coexistence of epilepsy and microcephaly, but do not differentiate microcephaly into relative and absolute [29,30]. For example, epilepsy, microcephaly, and short stature coexist in mutations in the glutaminyl-tRNA synthetase gene [31], and epilepsy, microcephaly, and normal height in Pitt–Hopkins syndrome [32].

In relative microcephaly we have both reduced hc (hc < x − 3s/< x − 2s) and h, therefore the condition HCI < x − 3s/< x − 2s is not met. In absolute microcephaly we found that a reduced hc (hc < x − 3s/< x − 2s) and the HCI condition < x − 3s/<x − 2s is fulfilled. It has been demonstrated that the higher the HCI value, the more disturbed the differentiation of body proportions, and the lower the chance of microcephaly. Microcephaly and short stature coexist, for example, not only in genetically determined syndromes, such as for example CdLS [33,34], Seckel syndrome [35]^,^ Rubinstein–Taybi syndrome [36], DS [37,38], ES [39], Cri du Chat syndrome [40], Emanuel syndrome [41], 7p22.1 microdeletions [42], but also in CP13 [43,44,45]. In our research, relative microcephaly defined on the basis of dysmorphological criteria occurred statistically significantly more often with CP, and on the basis of traditional criteria—with DS (with GD). In children and adolescents with CP, relative microcephaly was more severe (hc < x − 3s) than in children and adolescents with DS (hc < x − 2s). DS and CP are an example of NPE [8]. Absolute microcephaly coexisted more often with the mixed form in the CP subgroup, and significantly less frequently with the spastic form in the CP subgroup, regardless of the adopted criteria. Relative microcephaly (traditional criteria) coexisted more often with tetraplegia in the spastic subgroup, but statistical significance was not observed. The mixed form of CP is distinguished from the other forms of CP classified by Hagber. The spastic type is manifested by the pyramidal syndrome and results from damage to the white and/or gray matter of the brain hemispheres, the dyskinetic type is manifested by extrapyramidal syndrome and results from damage to the subcortical nuclei of the brain, the atactic type is manifested by the cerebellar syndrome and results from damage to the cerebellum, and the mixed type is manifested by the coexistence of at least two of the above-mentioned complexes of symptoms resulting from coexisting injuries of at least two parts of the brain [8,46].

NTDs [8] are also an example of NPE. It was shown that relative microcephaly defined on the basis of traditional criteria and absolute microcephaly defined on the basis of dysmorphological criteria were statistically and significantly less frequent with NTDs, and that relative microcephaly (both defined by dysmorphological and traditional criteria) coexisted significantly less frequently with sasMMC and HCP. In fact, microcephaly is not specific to patients with MMC sequential syndrome [47]. The presented NTDs with multifactorial inheritance should be distinguished from genetically determined congenital abnormalities: gene mutations [48] or chromosomal aberrations [49]. For example, in Meckel–Gruber syndrome, microcephaly is accompanied by occipital encephalocele [48]. Relative microcephaly, both defined on the basis of dysmorphological and traditional criteria, and absolute microcephaly defined on the basis of dysmorphological criteria, occurred significantly less frequently with NMD. It should be noted that there are syndromes of congenital malformations with classic, merizine-positive congenital dystrophy and microcephaly. An example is Fukuyama’s congenital muscular dystrophy [50].

In relative macrocephaly we have both increased hc (hc > x − 3s/> x −2s) and h, therefore the condition HCI >x − 3s/> x − 2s is not met. In the absolute macrocephaly we only have an increased hc (hc > x − 3s/> x − 2s) and the condition HCI > x − 3s/> x − 2s is fulfilled. It has been demonstrated that the higher the values within the HCI index, the more disturbed the differentiation of body proportions, the greater the chance of the occurrence of a macrocephaly. Taking into account the statistical significance, relative macrocephaly (defined on the basis of dysmorphological and traditional criteria) occurred significantly more often with PMS. Some authors note that normal stature and normal head size distinguish PMS syndrome from other genetic disorders [51], others emphasize that the presence of macrocephaly and tall stature depend on the extent of the deletion [52,53]. Macrocephaly and tall stature also coexist in Sotosa’s syndrome [54], in some connective tissue diseases [47,55], XYY syndrome [56], in the case of defects (truncating variants) in the chromodomain helicase DNA-binding protein 8 [57], large chromosomal deletions from 13q13.3 to 13q21.3 [58], Primrose syndrome [59], and Malan syndrome [60].

Relative macrocephaly defined on the basis of dysmorphological criteria occurred significantly more often with ACM. Absolute macrocephaly defined on the basis of dysmorphological and traditional criteria coexisted significantly more often with the NTDs subgroup, as well as with cases belonging the sasMMC and HCP and HCP subgroups. Absolute macrocephaly defined on the basis of traditional criteria coexisted with sasMMC. Macrocephaly occurred in sasMMC and HCP was more severe (hc > x − 3s) than in sasMMC (hc > x − 2s). The vast majority of patients with NTDs have hydrocephalus [61]. In infancy, before accretion of sutures and fontanel, the symptoms of intracranial hypertension associated with hydrocephalus are accompanied by a progressive macrocephalia [17]. In sasMMC and HCP short stature is attributed to smaller lower limbs [18,62], spinal deformities and scoliosis [63]. Macrocephaly at birth and higher levels of the defect in patients with sequential syndrome of MMC indicate a lower efficacy of neurosurgery treatment [47].

It was found that the following coexisted significantly more frequently: relative macrocephaly (defined on the basis of dysmorphological criteria) with DMD, and defined on the basis of traditional criteria with LGMD, moreover absolute macrocephaly (dysmorphological criteria) with HMSN and BMD, absolute macrocephaly (traditional criteria) with HMSN, BMD, and DMD. Other researchers have shown that macrocephaly occurs in conjunction with congenital muscular dystrophy due to mutation of gene laminin subunit alpha 2 [63], multifocal demyelinating motor neuropathy and Hamartoma syndrome associated with a ‘de novo’ mutation in the phosphatase and tensin homolog gene [64].

Relative macrocephaly (dysmorphological criteria) and absolute macrocephaly (dysmorphological and traditional criteria) coexist significantly less frequently with CP in the entire study group. Nevertheless, in the CP subgroup, the following coexisted significantly more often: relative macrocephaly (traditional criteria) with the atactic form, absolute macrocephaly (dysmorphological criteria) with the spastic form, and absolute macrocephaly (dysmorphological criteria) coexisted rarely with the mixed form of CP.

The study showed that among children and adolescents with neurodysfunction there were statistically significant relationships between short stature and spastic CP among patients with CP^13^. In the related literature, cases of spastic CP and macrocephaly are discussed [45]. Intraventricular haemorrhage is a frequent complication in extreme preterm births and is a risk factor for CP development [65,66]. High intraventricular haemorrhage grade, low gestational age at birth and increased head circumference were risk factors for post hemorrhagic hydrocephalus [65].

The dysmorphological and traditional criteria have different substantive meanings. Taking into account hc, abnormal head size defined on the basis of dysmorphological criteria is more advanced (more severe microcephaly/macrocephaly) than on the basis of traditional criteria. When we take into account the differentiation of body proportions expressed by the HCI value, we observe a change in the nomenclature depending on the criteria used, for example, relative macrocephaly meeting dysmorphological criteria, may also meet the traditional criteria of relative macrocephaly (macrocephaly and height stature coexist, body proportions are preserved) or absolute (macrocephaly occurs and normal height/short stature, body proportions are disturbed), and this can be observed in the example of DMD. The dysmorphological classification better differentiated the relationship between AHS and individual diseases/syndromes/separate subgroups (in all cases Cp was higher for the classification based on dysmorphological criteria than the traditional ones). On the other hand, application of the traditional classification made it possible to demonstrate a statistically significant relationship between the more frequent co-occurrence of absolute microcephaly and epilepsy. The knowledge of such similar dependencies enables close monitoring of children with isolated microcephaly for possible epileptic seizures [67]. Relative microcephaly is the most common, but in patients with absolute microcephaly there is a risk of epilepsy. Relative microcephaly (traditional criteria) rarely coexists with a low level of disability. The higher hc, HCI—the lower the level of GMFCS. It is known that the 4th and 5th level of GMFCS is more common in children with the tetraplegia (CP) [23].

Relative microcephaly and hypothyroidism coexisted more frequently in the study group, but statistical significance was not achieved. Thyroid hormones are crucial for the proper development of a child from the early stages of fetal life. They affect the development of the central nervous system in the prenatal period and, up to 3 years of age, regulate the child’s growth process and most metabolic processes [68]. More than half of premature babies who develop CP have proven (trans-parietal ultrasound) brain damage. The high prevalence of CP may be partly due to insufficient levels of neuroprotective substances such as thyroid hormone and glucocorticoids [69]. It is worth noting that in studies on the aetiology of microcephaly, classifications, taking into account hc, based on dysmorphological [28] or traditional criteria1 are used [70,71]. They are not used simultaneously, which makes it difficult to compare the results. Scientists and clinical practitioners should use both types of criteria when assessing abnormal head size because of their different connotations.

### 4.1. Clinical Implications

Learning about the above relationships is of practical importance, may facilitate the planning of diagnostic procedures and link the symptom (abnormal head size) with the main diagnosis or with a group of diseases that usually run with/without encephalopathy.

The most important clinical implications are as follows:relative microcephaly (regardless of the adopted criteria) often coexisted with NPE, and rarely with NMD in the entire study group,absolute microcephaly (regardless of the adopted criteria) often coexisted with the mixed form, and less frequently with spastic in the CP subgroup,relative macrocephaly (regardless of the adopted criteria) often occurred in PMS in the entire study group,absolute macrocephaly (regardless of the adopted criteria) commonly coexisted with NTDs, sasMMC and HCP, HCP, NMD, HMSN, and BMD, rarely with CP in the whole study group,absolute microcephaly (traditional criteria) commonly coexisted with epilepsy in the whole study group—relative microcephaly is the most common, but it is in patients with absolute microcephaly that there is a risk of epilepsy, while relative microcephaly (traditional criteria) is associated with a prognosis for a more severe level of disability,the application of both types of criteria, dysmorphological and traditional, has an important but different substantive meaning—both criteria should be used when assessing head size disturbances in children and adolescents.

### 4.2. Limitations

The study was retrospective and the data from medical records were scarce. From a clinical point of view, there are also other factors that significantly affect developmental disorders. For example, hormonal balances and head imaging studies should be taken into account in future prospective studies.

## 5. Conclusions

The presented definitions allowed demonstration of the differentiation in abnormal head size in children and adolescents with congenital nervous system disorders or neurological syndromes with one or more neurodysfunction visible since infancy. The abnormal head size in this group of children is most often represented by relative microcephaly.

The relationship has been found between relative and absolute microcephaly, relative and absolute macrocephaly and units or disease syndromes with neurodysfunction in a group of children, based on the data collected during the admission procedure to the Department of Children and Youth Neurological Rehabilitation. On this basis, clinical implications are presented.

The dysmorphological classification better differentiated the relationship between abnormal head size and individual diseases/syndromes or distinguished subgroups. The use of traditional criteria made it possible to demonstrate the relationship between the coexistence of absolute microcephaly and epilepsy. Both criteria should be used when assessing head size disturbances in children and adolescents.

## Figures and Tables

**Figure 1 jcm-09-03739-f001:**
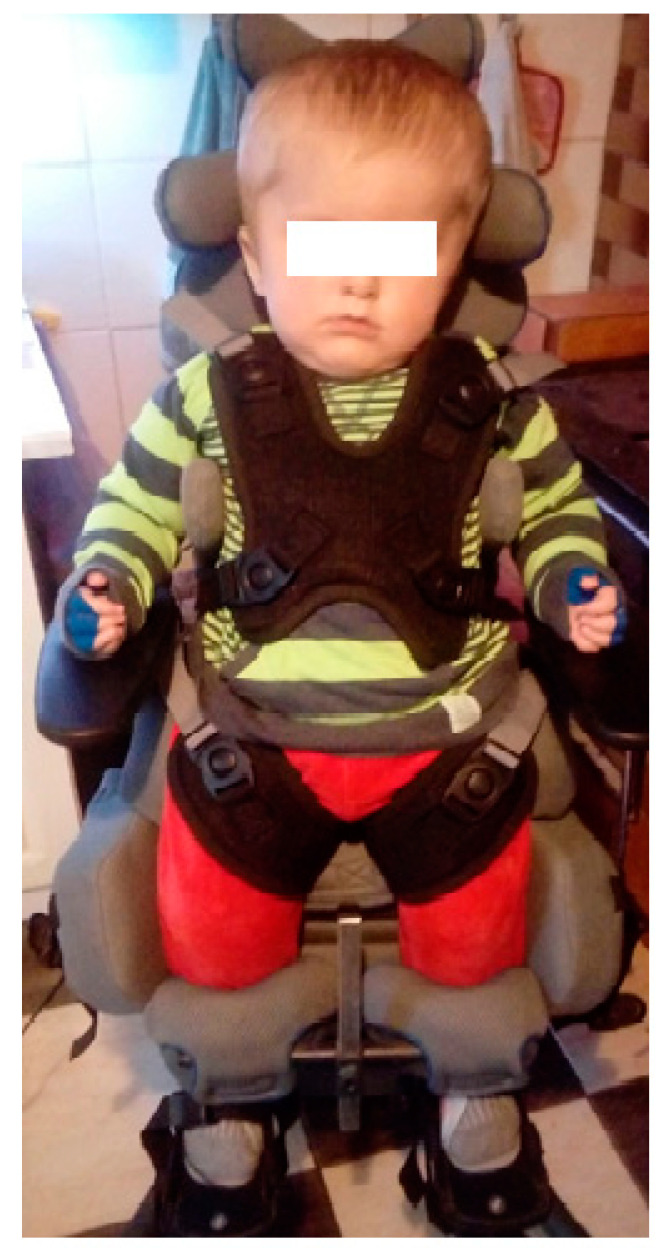
Boy 2 years 10 months: GMFCS V, condition after surgical treatment for congenital hydrocephalus, with epilepsy, weight: 16.8 kg, body high: 94 cm, head circumference: 55.5 cm (absolute macrocephaly). It would seem that he has been diagnosed and has a definitive diagnosis. Additional tests showed an increased level of creatine kinase: 4850 U/L [normative values: 68–293 U/L]—another level determination, elevated values in previous determinations. There are no diagnostics for neuromuscular diseases. Clinical problems: no obvious link between absolute macrocephaly and neuromuscular disease, no definitive diagnosis: one disease, coexistence of syndromes? There is a lack of reliable genetic counseling for parents.

**Figure 2 jcm-09-03739-f002:**
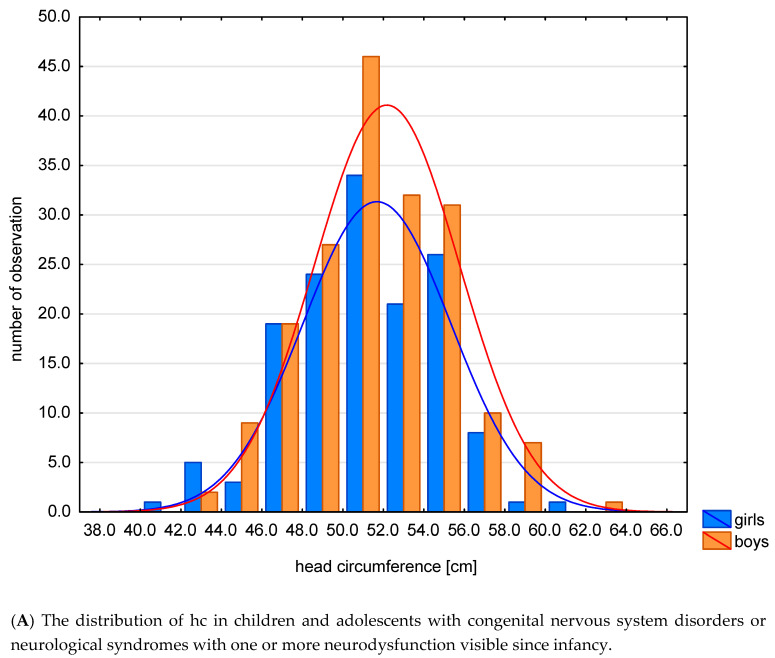
The distribution of hc (**A**) and HCI (**B**) in this specific population.

**Table 1 jcm-09-03739-t001:** Characteristics of diagnoses, subgroups, number of respondents (*N*), their percentage (*N*%), abbreviations used (**A**), and GMFCS score (**B**–**D**).

A. Study Group—Recognition, Division into Subgroups, Abbreviation											
Units and Syndromes Running with Neurodysfunction (Main Recognition)					Classification with Regard to Etiopathogenesis, Presence and Character Encephalopathy				Classification with Regard to Presence and Character Encephalopathy		
			*N*	*N*%			*N*	*N*%		*N*	*N*%
NBIA-MPAN, Neurodegeneration with Brain Iron Accumulation—Mitochondrial Protein Associated Neurodegeneration			2	0.6	MD, encephalopathy in metabolic disorder		7	2.1	PE, progressive encephalopathy	8	2.4
GSD II, Pompe’s disease			1	0.3							
LCHAD, long-chain 3-hydroxyacyl-coenzyme A dehydrogenase deficiency			1	0.3							
SLO, Smith-Lemli-Opitz syndrome			1	0.3							
GLUT1d, glucose transporter 1 deficiency			1	0.3							
NKH, nonketotic hyperglycinemia			1	0.3							
SMEI, Dravet’s syndrome			1	0.3	EE, epileptic encephalopathy		1	0.3			
sasMMC and HCP, state after surgery lumbar myelomeningocele and hydrocephalus			17	5.2	NTDs, encephalopathy in neural tube defects		24	7.3	NPE, non progressive encephalopathy	287	88.1
sasMMC, state after surgery lumbar myelomeningocele			3	0.9							
sasMM, state after surgery parietooocipital menigocele			1	0.3							
ACM, Arnold-Chiari malformation			2	0.6							
HCP, isolated hydrocephalus			1	0.3							
DS, Down syndrome			11	3.4	GD, encephalopathy in genetic disorders		23	7.0			
ES, Edwards syndrome			1	0.3							
PMS, Phelan-McDermid syndrome			2	0.6							
MWS, Mowat-Wilson syndrome			1	0.3							
AS, Angelman syndrome			1	0.3							
DGS, Di George syndrome			1	0.3							
46,XY,del(X)(q24)			1	0.3							
CdLS, Cornelia de Lange syndrome			1	0.3							
SDS, Schwachman-Diamond syndrome			1	0.3							
PWS, Prader-Willi syndrome			1	0.3							
46 XX, add(2)(q25)			1	0.3							
46XX, del (12) (q24.21q24.23)			1	0.3							
FAS, fetal alcohol syndrome			1	0.3	TE, toxic encephalopathy		1	0.3			
CP, cerebral palsy			239	73.1	CP encephalopathy in cerebral palsy		239	73.1			
HMSN, hereditary motor and sensory polyneuropathy			8	2.4	NMD, neuromuscular disorders		32	9.8	NMD, neuromuscular disorders	32	9.8
LGMD, muscular dystrophy limb-girdle			7	2.1							
BMD, Becker’s muscular dystrophy			3	0.9							
DMD, Duchenne muscular dystrophy			7	2.1							
TD, Thomsen disease			1	0.3							
AMC&N arthrogryposis multiplet congenita with neuropathy			3	0.9							
CM, congenital myopathy			1	0.3							
SMA, spinal muscular atrophy			2	0.6							
In total			327	100	In total		327	100	In total	327	100
**B. The Level of GMFC—The Greater the Number of Points, The Greater the Level of Motor Disability**											
GMFCS I-V		I	II		III		IV		V		
Number of points assigned		1	2		3		4		5		
GMFCS A-C		A			B		C				
Number of points assigned		1			2		3				
**C. The Statistical Characteristics of the GMFCS Score in the Entire Study Group**											
Parameters	*N*	$\bar{x}$	Me		*s*	*c* _25_	*c* _75_		Min	Max	
GMFCS I-V	327	2.47	2		1.29	2	3		1	5	
GMFCS A-C		1.60	1		0.86	1	2		1	3	
**D. The Statistical Characteristics of the GMFCS Score in the Subgroup with CP**											
Parameters	*N*	$\bar{x}$	Me		*s*	*c* _25_	*c* _75_		Min	Max	
GMFCS I-V	239	2.48	2		1.33	1	3.5		1	5	
GMFCS A-C		1.62	1		0.86	1	2.5		1	3	

Gross Motor Function Classification System (GMFCS), arithmetic mean ($\bar{x}$), median (Me), standard deviation (s), smallest (Min) and largest value (Max), 25th centile (*c_25_*) and 75th (*c_75_*).

**Table 2 jcm-09-03739-t002:** Quantitative characteristic: head circumference (hc), h, head circumference index (HCI), z-score hc, z-score h, z-score HCI (**A**), developmental disorders of head size (**B**–**E**), developmental disorders of the head size and z-score hc/h (**F**).

A. Statistical Characteristics of Indicators: z-Score hc, z-Score h, z-Score HCI									
Parameters		*N*	$\bar{x}$	Me	*s*	*c* _25_	*c* _75_	Min	Max
hc		327	51.89	52	3.60	50	54.75	42.1	63.3
h			130.78	126	24.32	110	150.50	83.9	191.2
hc/h			0.41	0.40	0.06	0.36	0.46	0.29	0.61
z-score hc			−0.53	−0.54	2.14	−1.71	0.86	−7.36	8.29
z-score h			−1.23	−1.16	1.98	−2.33	−0.05	−8.93	4.20
z-score hc/h			0.90	0.85	2.04	−0.44	2.20	−4.38	11.29
arithmetic mean ($\bar{x}$), median (Me), standard deviation (s), smallest (Min) and largest value (Max), 25th centile (*c_25_*) and 75th (*c_75_*)									
**B. The Size of the Head: Dysmorphology and Traditional Classification (hc)**									
**The Size of the Head**			**Dysmorphology Classification (hc)**				**Traditional Classification (hc)**		
Normal			−3 ≥ z-score hc ≤ 3				−2 ≥ z-score hc ≤ 2		
Microcephaly			z-score hc < −3				z-score hc < −2		
Macrocephaly			z-score hc > 3				z-score hc > 2		
**C. The Abnormal Size of the Head: Dysmorphology and Traditional Classification (hc and HCI)**									
**The Abnormal Size of the Head**			**Dysmorphology Classification (hc and HCI)**				**Traditional Classification (hc and HCI)**		
			**hc**	**HCI**			**hc**	**HCI**	
Relative microcephaly			z-score hc<−3	z-score HCI ≠ (<−3)			z-score hc <−2	z-score HCI ≠ (<−2)	
Absolute microcephaly				z-score HCI < −3				z-score HCI<−2	
Relative macrocephaly			z-score hc>3	z-score HCI ≠ (> 3)			z-score hc > 2	z-score HCI ≠ (>2)	
Absolute macrocephaly				z-score HCI > 3				z-score HCI > 2	
**D. The Incidence of Abnormal Head Size Defined Based on Dysmorphology Classification (hc and HCI)**									
**The Size of the Head—Dysmorphology Classification (hc)**						**The Abnormal Size of the Head—Dysmorphology Classification (hc and HCI)**			
Normal	273	83.5	Normal	273	83.5	83.5			
Abnormal	54	16.5	Microcephaly	41	12.5	11.0	Relative microcephaly	36	66.7
						1.5	Absolute microcephaly	5	9.3
			Macrocephaly	13	4	1.5	Relative macrocephaly	5	9.3
						2.5	Absolute macrocephaly	8	14.8
In total	327	100.0	In total	327	100.0	100.0	In total	54	100.0
	N	*N*%		N	*N*%	*N*%		N	*N*%
**E. The Incidence of Abnormal Head Size Defined based on Traditional Classification (hc and HCI)**									
**The Size of the Head—Traditional Classification (hc)**							**The Abnormal Size of the Head—Traditional Classification (hc and HCI)**		
Normal	224	68.5	Normal	224	68.5	68.5			
Abnormal	103	31.5	Microcephaly	72	22.0	18.3	Relative microcephaly	60	58.3
						3.7	Absolute microcephaly	12	11.7
			Macrocephaly	31	9.5	4.3	Relative macrocephaly	14	13.6
						5.2	Absolute macrocephaly	17	16.5
In total	327	100.0	In total	327	100.0	100.0	In total	103	100.0
	N	*N*%		N	*N*%	*N*%		N	*N*%
**F. Normal Head Size, Microcephaly, Macrocephaly and z-score HCI**									
**Nominal Regression**			Quantitative dependent variable z-score HCI			Quantitative dependent variable z-score HCI	**Nominal Regression**		
Qualitative dependent variable The size of the head, dysmorphology classification (hc)	Normal 273(83.5%)						Normal 224 (68.5%)	Qualitative dependent variable The size of the head, traditional classification (hc)	
	Microcephaly 41(12.5%)		0.001		*p*	0.028	Microcephaly 72 (22.0%)		
			0.743 (0.620–0.890)		OR	0.854 (0.742–0.983)			
	Macrocephaly 13(4.0%)		0.004		*p*	0.006	Macrocephaly 31 (9.5%)		
			1.446 (1.124–1.860)		OR	1.291 (1.076–1.548)			

hc—head circumference, h—body height, HCI—head circumference index, N—numbers of patients, %—percent, *p*—probability value calculated by chi-square test of independence, OR—Odds Ratio (95% confidence interval).

**Table 3 jcm-09-03739-t003:** The abnormal size of the head and units and syndromes running with neurodysfunction (**A**,**B**) and classification with regard to etiopathogenesis, presence and character of encephalopathy (**C**).

Units and Syndromes Running with Neurodysfunction	A. The Abnormal Size of the Head—Dysmorphology Classification (hc and HCI) by z-Score hc and HCI (*p* < 0.001; Cp = 0.736)								
	Relative Microcephaly		Absolute Microcephaly		Relative Macrocephaly		Absolute Macrocephaly		In Total
	*N* (*N*%)	ASR	*N* (*N*%)	ASR	*N* (*N*%)	ASR	*N* (*N*%)	ASR	
sasMMC and HCP	0 (0.0%)	−2.0	0 (0.0%)	−0.5	0 (0.0%)	−0.5	2 (100.0%)	3.5	2 (100.0%)
ACM	0 (0.0%)	−1.4	0 (0.0%)	−0.3	1 (100.0%)	3.2	0 (0.0%)	−0.4	1 (100.0%)
HCP	0 (0.0%)	−1.4	0 (0.0%)	−0.3	0 (0.0%)	−0.3	1 (100.0%)	2.4	1 (100.0%)
DS	3 (100.0%)	1.3	0 (0.0%)	−0.6	0 (0.0%)	−0.6	0 (0.0%)	−0.7	3 (100.0%)
ES	1 (100.0%)	0.7	0 (0.0%)	−0.3	0 (0.0%)	−0.3	0 (0.0%)	−0.4	1 (100.0%)
PMS	0 (0.0%)	−1.4	0 (0.0%)	−0.5	1 (100.0%)	3.2	0 (0.0%)	−0.4	1 (100.0%)
DGS	1 (100.0%)	0.7	0 (0.0%)	−0.3	0 (0.0%)	−0.3	0 (0.0%)	−0.4	1 (100.0%)
CP	31 (75.6%)	2.5	5 (12.2%)	1.3	2 (4.9%)	−2.0	3 (7.3%)	−2.8	41 (100.0%)
HMSN	0 (0.0%)	−1.4	0 (0.0%)	−0.3	0 (0.0%)	−0.3	1 (100.0%)	2.4	1 (100.0%)
BMD	0 (0.0%)	−1.4	0 (0.0%)	−0.3	0 (0.0%)	−0.3	1 (100.0%)	2.4	1 (100.0%)
DMD	0 (0.0%)	−1.4	0 (0.0%)	−0.3	1 (100.0%)	3.2	0 (0.0%)	−0.4	1 (100.0%)
In total	36 (66.7%)		5 (9.3%)		5 (9.3%)		8 (14.3%)		54 (100.0%)
**Units and Syndromes Running with Neurodysfunction**	**B. The Abnormal Size of the Head—Traditional Classification (hc and HCI) by z-score hc and HCI (*p* = 0.002; Cp = 0.679)**								
	**Relative** **Microcephaly**		**Absolute** **Microcephaly**		**Relative** **Macrocephaly**		**Absolute** **Macrocephaly**		**In Total**
	***N*** **(*N*%)**	**ASR**	***N*** **(*N*%)**	**ASR**	***N*** **(*N*%)**	**ASR**	***N*** **(*N*%)**	**ASR**	
SLO	1 (100.0%)	0.9	0 (0.0%)	−0.4	0 (0.0%)	−0.4	0 (0.0%)	−0.4	1 (100.0%)
sasMMC and HCP	0 (0.0%)	−2.1	0 (0.0%)	−0.6	0 (0.0%)	−0.7	3 (100.0%)	4.0	3 (100.0%)
sasMMC	0 (0.0%)	−1.7	0 (0.0%)	−0.5	0 (0.0%)	−0.6	2 (100.0%)	3.2	2 (100.0%)
MM	1 (100.0%)	0.9	0 (0.0%)	−0.4	0 (0.0%)	−0.4	0 (0.0%)	−0.4	1 (100.0%)
ACM	1 (50.0%)	−0.2	0 (0.0%)	−0.5	1 (50.0%)	1.5	0 (0.0%)	−0.6	2 (100.0%)
HCP	0 (0.0%)	−1.2	0 (0.0%)	−0.4	0 (0.0%)	−0.4	1 (100.0%)	2.3	1 (100.0%)
DS	6 (100.0%)	2.1	0 (0.0%)	−0.9	0 (0.0%)	−1.0	0 (0.0%)	−1.1	6 (100.0%)
ES	1 (100.0%)	0.9	0 (0.0%)	−0.4	0 (0.0%)	−0.4	0 (0.0%)	−0.4	1 (100.0%)
PMS	0 (0.0%)	−1.2	0 (0.0%)	−0.4	1 (100.0%)	2.5	0 (0.0%)	−0.4	1 (100.0%)
AS	1 (100.0%)	0.9	0 (0.0%)	−0.4	0 (0.0%)	−0.4	0 (0.0%)	−0.4	1 (100.0%)
DGS	1 (100.0%)	0.9	0 (0.0%)	−0.4	0 (0.0%)	−0.4	0 (0.0%)	−0.4	1 (100.0%)
FAS	1 (100.0%)	0.9	0 (0.0%)	−0.4	0 (0.0%)	−0.4	0 (0.0%)	−0.4	1 (100.0%)
CP	45 (62.5%)	1.3	12 (16.7%)	2.4	8 (11.1%)	−1.1	7 (9.7%)	−2.8	72 (100.0%)
HMSN	0(0.0%)	−1.2	0 (0.0%)	−0.4	0 (0.0%)	−0.4	1 (100.0%)	2.3	1 (100.0%)
LGMD	0 (0.0%)	−2.7	0 (0.0%)	−0.8	4 (80.0%)	4.4	1 (20.0%)	0.2	5 (100.0%)
BMD	0 (0.0%)	−1.2	0 (0.0%)	−0.4	0 (0.0%)	−0.4	1 (100.0%)	2.3	1 (100.0%)
DMD	0 (0.0%)	−1.2	0 (0.0%)	−0.4	0 (0.0%)	−0.4	1 (100.0%)	2.3	1 (100.0%)
AMC and N	1 (100.0%)	0.9	0 (0.0%)	−0.4	0 (0.0%)	−0.4	0 (0.0%)	−0.4	1 (100.0%)
SMA	1 (100.0%)	0.9	0 (0.0%)	−0.4	0 (0.0%)	−0.4	0 (0.0%)	−0.4	1 (100.0%)
In total	60 (58.3%)		12 (11.7%)		14 (13.6%)		17 (16.5%)		103 (100.0%)
**Classification with Regard to Etiopathogenesis, Presence and Character Encephalopathy**	**C. The Abnormal Size of the Head—Dysmorphology Classification (hc and HCI by z-score hc and HCI (*p* = 0.001; Cp = 0.590)**								
	**Relative** **Microcephaly**		**Absolute** **Microcephaly**		**Relative** **Macrocephaly**		**Absolute** **Macrocephaly**		**In Total**
	***N*** **(*N*%)**	**ASR**	***N*** **(*N*%)**	**ASR**	***N*** **(*N*%)**	**ASR**	***N*** **(*N*%)**	**ASR**	
NTDs	0 (0.0%)	−2.9	0 (0.0%)	−0.7	1 (25.0%)	1.1	3 (75.0%)	3.5	4 (100.0%)
GD	5 (83.3%)	0.9	0 (0.0%)	−0.8	1 (16.7%)	0.7	0 (0.0%)	-1.1	6 (100.0%)
CP	31 (75.6%)	2.5	5 (12.2%)	1.3	2 (4.9%)	−2.0	3 (7.3%)	−2.8	41 (100.0%)
NMD	0 (0.0%)	−2.5	0 (0.0%)	−0.6	1 (33.3%)	1.5	2 (66.7%)	2.6	3 (100.0%)
In total	36 (66.7%)		5 (9.3%)		5 (9.3%)		8 (14.3%)		54 (100.0%)

*N*—numbers of patients, %—percent, *p*—probability value calculated by chi-square test of independence, Cp—Pearson’s Contingency Coefficient C, Cp ≥ 0, when is distant from 0, there is some relationship and the closer to 1, a perfect association will approach, ASR—Adjusted Standardized Residuals, values >1.96 reflect a greater number, and those below <−1.96 correspond to a smaller number than random distribution.

**Table 4 jcm-09-03739-t004:** Abnormal size of the head and classification with regard to etiopathogenesis, presence and character encephalopathy (**A**)/and classification with regard to presence and character encephalopathy (**B**,**C**), abnormal head size and types of CP (**D**,**E**).

Classification with Regard to Etiopathogenesis, Presence and Character Encephalopathy	A. The Abnormal Size of the Head—Traditional Classification (hc and HCI) by z-Score hc and HCI (*p* = 0.001; Cp = 0.525)								
	Relative Microcephaly		Absolute Microcephaly		Relative Macrocephaly		Absolute Macrocephaly		In Total
	*N* (*N*%)	ASR	*N* (*N*%)	ASR	*N* (*N*%)	ASR	*N* (*N*%)	ASR	
MD	1 (100.0%)	0.9	0 (0.0%)	−0.4	0 (0.0%)	−0.4	0 (0.0%)	−0.4	1 (100.0%)
NTDs	2 (22.2%)	−2.3	0 (0.0%)	−1.1	1 (11.1%)	−0.2	6 (66.7%)	4.2	9 (100.0%)
GD	9 (90.0%)	2.1	0 (0.0%)	−1.2	1 (10.0%)	−0.3	0 (0.0%)	−1.5	10 (100.0%)
TE	1 (100.0%)	0.9	0 (0.0%)	−0.4	0 (0.0%)	−0.4	0 (0.0%)	−0.4	1 (100.0%)
CP	45 (62.5%)	1.3	12 (16.7%)	2.4	8 (11.1%)	−1.1	7 (9.7%)	−2.8	72 (100.0%)
NMD	2 (20.0%)	−2.6	0 (0.0%)	−1.2	4 (40.0%)	2.6	4 (40.0%)	2.1	10 (100.0%)
In total	60 (58.3%)		12 (11.7%)		14 (13.6%)		17 (16.5%)		103 (100.0%)
**Classification with Regard to Presence and Character Encephalopathy**	**B. The Abnormal Size of the Head—Dysmorphology Classification (hc and HCI) by z−Score hc and HCI (*p* = 0.001; Cp = 0.590)**								
	**Relative** **Microcephaly**		**Absolute** **Microcephaly**		**Relative** **Macrocephaly**		**Absolute** **Macrocephaly**		**In Total**
	***N* (*N*%)**	**ASR**	***N* (*N*%)**	**ASR**	***N* (*N*%)**	**ASR**	***N* (*N*%)**	**ASR**	
NPE	36 (70.6%)	2.5	5 (9.8%)	0.6	4 (7.8%)	−1.5	6 (11.8%)	−2.6	51 (100.0%)
NMD	0 (0.0%)	−2.5	0 (0.0%)	−0.6	1 (33.3%)	1.5	2 (66.7%)	2.6	3 (100.0%)
In total	36 (66.7%)		5 (9.3%)		5 (9.3%)		8 (14.3%)		54 (100.0%)
**Classification with Regard to Presence and Character Encephalopathy**	**C. The Abnormal Size of the head—Traditional Classification (hc and HCI) by z-Score hc and HCI (*p* = 0.029; Cp = 0.347)**								
	**Relative** **Microcephaly**		**Absolute** **Microcephaly**		**Relative** **Macrocephaly**		**Absolute** **Macrocephaly**		**In Total**
	***N* (*N*%)**	**ASR**	***N* (*N*%)**	**ASR**	***N* (*N*%)**	**ASR**	***N* (*N*%)**	**ASR**	
PE	1 (100.0%)	0.9	0 (0.0%)	−0.4	0 (0.0%)	−0.4	0 (0.0%)	−0.4	1 (100.0%)
NPE	57 (62.0%)	2.2	12 (13.0%)	1.3	10 (10.9%)	−2.3	13 (14.1%)	−1.9	92 (100.0%)
NMD	0 (20.0%)	−2.6	0 (0.0%)	−1.2	4 (40.0%)	2.6	4 (40.0%)	2.1	10 (100.0%)
In total	60 (58.3%)		12 (11.7%)		14 (13.6%)		17 (16.5%)		103 (100.0%)
**Types of CP**	**D. The Abnormal Size of the Head—Dysmorphology Classification (hc and HCI) by z-Score hc and HCI (*p* = 0.029; Cp = 0.425)**								
	**Relative** **Microcephaly**		**Absolute** **Microcephaly**		**Relative** **Macrocephaly**		**Absolute** **Macrocephaly**		**In Total**
	***N* (*N*%)**	**ASR**	***N* (*N*%)**	**ASR**	***N* (*N*%)**	**ASR**	***N* (*N*%)**	**ASR**	
Spastic type	30 (78.9%)	1.8	3 (7.9%)	−3.0	2 (5.3%)	0.4	3 (7.9%)	3.5	38 (100.0%)
Mixed type	1 (33.3%)	−1.8	2 (66.7%)	3.0	0 (0.0%)	−0,4	0 (0.0%)	−2.8	3 (100.0%)
In total	31 (75.6%)		5 (12.2%)		2 (4.9%)		3 (7.3%)		41 (100.0%)
**Types of CP**	**E. The Abnormal Size of the Head—Traditional Classification (hc and HCI) by z−Score hc and HCI (*p* = 0.016; Cp = 0.422)**								
	**Relative** **Microcephaly**		**Absolute** **Microcephaly**		**Relative** **Macrocephaly**		**Absolute** **Macrocephaly**		**In Total**
	***N* (*N*%)**	**ASR**	***N* (*N*%)**	**ASR**	***N* (*N*%)**	**ASR**	***N* (*N*%)**	**ASR**	
Spastic type	43 (65.2%)	1.5	9 (13.6%)	−2.3	7 (10.6%)	−0.5	7 (10.6%)	0.8	66 (100.0%)
Atactic type	0 (0.0%)	−1.3	0 (0.0%)	−0.5	1 (100.0%)	2.8	0 (0.0%)	−0.3	1 (100.0%)
Mixed type	2 (40.0%)	−1.1	3 (60.0%)	2.7	0 (0.0%)	−0.8	0 (0.0%)	−0.8	5 (100.0%)
In total	45 (62.5%)		12 (16.7%)		8 (11.1%)		7 (9.7%)		72 (100.0%)

*N*—numbers of patients, %—percent, *p*—probability value calculated by chi-square test of independence, Cp—Pearson’s Contingency Coefficient C, Cp ≥ 0, when is distant from 0, there is some relationship and the closer to 1, a perfect association will approach, ASR—Adjusted Standardized Residuals, values >1.96 reflect a greater number, and those below <−1.96 correspond to a smaller number than random distribution.

**Table 5 jcm-09-03739-t005:** Abnormal head size and epilepsy (**A**), and other dependencies—abscence of statistically significant relationships (**B**–**F**).

Accompanying Recognition Epilepsy	A. The Abnormal Size of the Head—Traditional Classification (hc and HCI) by z-Score hc and HCI (*p* = 0.043; Cp = 0.271)								
	Relative Microcephaly		Absolute Microcephaly		Relative Macrocephaly		Absolute Macrocephaly		In Total
	*N* (*N*%)	ASR	*N* (*N*%)	ASR	*N* (*N*%)	ASR	*N* (*N*%)	ASR	
Present	26 (63.4%)	0.9	8 (19.5%)	2.0	4 (9.8%)	−0.9	3 (7.3%)	−2.0	41 (100.0%)
Lack	34 (54.8%)	−0.9	4 (6.5%)	−2.0	10 (16.1%)	0.9	14 (22.6%)	2.0	62 (100.0%)
In total	60 (58.3%)		12 (11.7%)		14 (13.6%)		17 (16.5%)		103 (100.0%)
**Kind of** **Spastic Type**	**B. The Abnormal Size of the Head—Traditional Classification (hc and HCI) by z-Score hc and HCI (*p* = 0.312; Cp = 0.311)**								
	**Relative** **Microcephaly**		**Absolute** **Microcephaly**		**Relative** **Macrocephaly**		**Absolute** **Macrocephaly**		**In Total**
	***N* (*N*%)**	**ASR**	***N* (*N*%)**	**ASR**	***N* (*N*%)**	**ASR**	***N* (*N*%)**	**ASR**	
Diplegia	16 (57.1%)	−1.2	3 (10.7%)	-0.6	4 (14.3%)	0.8	5 (17.9%)	1.6	28 (100.0%)
Hemiplegia	5 (50.0%)	−1.1	3(30.0%)	1.6	1 (10.0%)	−0.1	1 (10.0%)	−0.1	10 (100.0%)
Tertaplegia	22 (78.6%)	2.0	3 (10.7%)	−0.6	2 (7.1%)	−0.8	1 (3.6%)	−1.6	28 (100.0%)
In total	43 (65.2%)		9 (13.6%)		7 (10.6%)		7 (10.5%)		103 (100.0%)
**Accompanying Recognition** **Hypothyroidism**	**C. The Abnormal Size of The Head—Traditional Classification (Hc and Hci) By Z-Score Hc and Hci (*p* = 0.207; Cp = 0.206)**								
	**Relative** **Microcephaly**		**Absolute** **Microcephaly**		**Relative** **Macrocephaly**		**Absolute** **Macrocephaly**		**In Total**
	***N* (*N*%)**	**ASR**	***N* (*N*%)**	**ASR**	***N* (*N*%)**	**ASR**	***N* (*N*%)**	**ASR**	
Present	8 (88.9%)	2.0	1 (11.1%)	−0.1	0 (0.0%)	−1.2	0 (0.0%)	−1.4	9 (100.0%)
Lack	52 (55.3%)	-2.0	11 (11.7%)	0.1	14 (14.9%)	1.2	17 (18.1%)	1.4	94 (100.0%)
In total	60 (58.3%)		12 (11.7%)		14 (13.6%)		17 (16.5%)		103 (100.0%)
**Kind of Spastic Type**	**D. The Abnormal Size of the Head—Dysmorphological Classification (hc and HCI) by z-Score hc and HCI (*p* = 0.498; Cp = 0.352)**								
	**Relative** **Microcephaly**		**Absolute** **Microcephaly**		**Relative** **Macrocephaly**		**Absolute** **Macrocephaly**		**In Total**
	***N* (*N*%)**	**ASR**	***N* (*N*%)**	**ASR**	***N* (*N*%)**	**ASR**	***N* (*N*%)**	**ASR**	
Diplegia	9 (75.0%)	−0.4	0 (0.0%)	−1.2	1 (8.3%)	0.6	2 (16.7%)	1.4	12 (100.0%)
Hemiplegia	6 (75.0%)	−0.3	1 (12.5%)	0.5	1 (12.5%)	1.0	0 (0.0%)	−0.9	8 (100.0%)
Tertaplegia	15 (83.3%)	0.6	2 (11.1%)	0.7	0 (0.0%)	−1.4	1 (5.6%)	−0.5	18 (100.0%)
In total	30 (78.9%)		3 (7.9%)		2 (5.3%)		3 (7.9%)		38 (100.0%)
**Accompanying Recognition** **Epilepsy**	**E. The Abnormal Size of the Head—Dysmorphological Classification (hc and HCI) by z-Score hc and HCI (*p* = 0.416; Cp = 0.224)**								
	**Relative** **Microcephaly**		**Absolute** **Microcephaly**		**Relative** **Macrocephaly**		**Absolute** **Macrocephaly**		**In Total**
	***N* (*N*%)**	**ASR**	***N* (*N*%)**	**ASR**	***N* (*N*%)**	**ASR**	***N* (*N*%)**	**ASR**	
Present	20 (74.1%)	1.2	3 (11.1%)	0.5	2 (7.4%)	−0.5	2 (7.4%)	−1.5	27 (100.0%)
Lack	16 (59.3%)	−1.2	2 (7.4%)	−0.5	3 (11.1%)	0.5	6 (22.2%)	1.5	27 (100.0%)
In total	36 (66.7%)		5 (9.3%)		5 (9.3%)		8 (14.3%)		54 (100.0%)
**Accompanying recognition** **Hypothyroidism**	**F. The Abnormal Size of the Head—Dysmorphological Classification (hc and HCI) by z-Score hc and HCI (*p* = 0.431; Cp = 0.220)**								
	**Relative** **microcephaly**		**Absolute** **Microcephaly**		**Relative** **macrocephaly**		**Absolute** **macrocephaly**		**In total**
	***N* (*N*%)**	**ASR**	***N* (*N*%)**	**ASR**	***N* (*N*%)**	**ASR**	***N* (*N*%)**	**ASR**	
Present	5 (100.0%)	1.7	0 (0.0%)	−0.7	0 (0.0%)	−0.7	0 (0.0%)	−1.0	5 (100.0%)
Lack	31 (63.3%)	−1.7	5 (10.2%)	0.7	5 (10.2%)	0.7	8 (16.3%)	1.0	49 (100.0%)
In total	36 (66.7%)		5 (9.3%)		5 (9.3%)		8 (14.3%)		54(100.0%)

*N*—numbers of patients, %—percent, *p*—probability value calculated by chi-square test of independence, Cp—Pearson’s Contingency Coefficient C, Cp ≥ 0, when is distant from 0, there is some relationship and the closer to 1, a perfect association will approach, ASR—Adjusted Standardized Residuals, values >1.96 reflect a greater number, and those below <−1.96 correspond to a smaller number than random distribution.

**Table 6 jcm-09-03739-t006:** Abnormal head size and level of GMFCS.

A. Dependence Between Absolute Value of Anthropometric Characteristics and Level of GMFCS in the Entire Study Group									
Variable Pairs	R		*p*		Variable pairs		R		*p*
hc [cm] vs. GMFCS I–V	−0.28		0.000		HCI [cm/cm] vs. GMFCS I-V		−0.22		0.000
hc [cm] vs. GMFCS A–C	−0.17		0.002		HCI [cm/cm] vs. GMFCS A−C		−0.11		0.039
Spearman’s rank correlation, *p*—probability value, R—Spearman’s rank correlation coefficient									
**B. The Abnormal Size of the Head—Dysmorphological Classification (hc and HCI) and Level of GMFCS I–V in the Entire Study Group**									
***p* = 0.057** **Cp = 0.53**	**Relative Microcephaly** ***N* (*N*%)** **ASR**		**Absolute Microcephaly** ***N* (*N*%)** **ASR**		**Relative macrocephaly** ***N* (*N*%)** **ASR**		**Absolute macrocephaly** ***N* (*N*%)** **ASR**		**In Total**
GMFCS I	3 (42.9%)	−1.4	0 (0.0%)	−0.9	1 (14.3%)	0.5	3 (42.9%)	2.2	7 (100.0%)
GMFCS II	11 (61.1%)	−0.6	2 (11.1%)	0.3	3 (16.7%)	1.3	2 (11.1%)	−0.5	18 (100.0%)
GMFCS III	6 (100.0%)	1.8	0 (0.0%)	−0.8	0 (0.0%)	−0.8	0 (0.0%)	−1.1	6 (100.0%)
GMFCS IV	3 (42.9%)	−1.4	0 (0.0%)	−0.9	1 (14.3%)	0.5	3 (42.9%)	2.2	7 (100.0%)
GMFCS V	13 (81.2%)	1.5	3 (18.8%)	1.6	0 (0.0%)	−1.5	0 (0.0%)	−2	16 (100.0%)
In total	36 (66.7%)		5 (9.3%)		5 (9.3%)		8 (14.8%)		54 (100.0%)
**C. The Abnormal Size of the Head—Traditional Classification (hc and HCI) and Level of GMFCS I–V in the Entire Study Group**									
***p* = 0.055** **Cp = 0.41**	**Relative Microcephaly** ***N* (*N*%)** **ASR**		**Absolute Microcephaly** ***N* (*N*%)** **ASR**		**Relative Macrocephaly** ***N* (*N*%)** **ASR**		**Absolute Macrocephaly** ***N* (*N*%)** **ASR**		**In Total**
GMFCS I	10 (45.5%)	−1.4	1 (4.5%)	−1.2	5 (22.7%)	1.4	6 (27.3%)	1.5	22 (100.0%)
GMFCS II	18 (50.0%)	−1.2	4 (11.1%)	−0.1	8 (22.2%)	1.9	6 (16.7%)	0	36 (100.0)
GMFCS III	6 (85.7%)	1.5	1 (14.3%)	0.2	0 (0.0%)	−1.1	0 (0.0%)	−1.2	7 (100.0%)
GMFCS IV	7 (53.8%)	−0.3	1 (7.7%)	−0.5	1 (7.7%)	−0.7	4 (30.8%)	1.5	13 (100.0%)
GMFCS V	19 (76.0%)	2.1	5 (20.0%)	1.5	0 (0.0%)	−2.3	1 (4.0%)	−1.9	25 (100.0%)
In total	60 (58.3%)		12 (11.7%)		14 (13.6%)		17 (16.5%)		103 (100.0%)
**D. The Abnormal Size of the Head—Dysmorphological Classification (hc and HCI) and Level of GMFCS A–C in the Entire Study Group**									
***p* = 0.403** **Cp = 0.32**	**Relative Microcephaly** ***N* (*N*%)** **ASR**		**Absolute Microcephaly** ***N* (*N*%)** **ASR**		**Relative Macrocephaly** ***N* (*N*%)** **ASR**		**Absolute macrocephaly** ***N* (*N*%)** **ASR**		**In Total**
GMFCS A	14 (56.0%)	−1.5	2 (8.0%)	−0.3	4 (16.0%)	1.6	5 (20.0%)	1	25 (100.0%)
GMFCS B	6 (100.0%)	1.8	0 (0.0%)	−0.8	0 (0.0%)	−0.8	0 (0.0%)	−1.1	6 (100.0%)
GMFCS C	16 (69.6%)	0.4	3 (13.0%)	0.8	1 (4.3%)	−1.1	3 (13.0%)	−0.3	23 (100.0%)
In total	36 (66.7%)		5 (9.3%)		5 (9.3%)		8 (14.8%)		54 (100.0%)
**E. The Abnormal Size of the Head—Traditional Classification (hc and HCI) and Level of GMFCS A–C in the Entire Study Group**									
***p* = 0.039** **Cp = 0.34**	**Relative Microcephaly** ***N* (*N*%)** **ASR**		**Absolute Microcephaly** ***N* (*N*%)** **ASR**		**Relative Macrocephaly** ***N* (*N*%)** **ASR**		**Absolute Macrocephaly** ***N* (*N*%)** **ASR**		**In Total**
GMFCS A	28 (48.3%)	–2.3	5 (8.6%)	−1.1	13 (22.4%)	3	12 (20.7%)	1.3	58 (100.0%)
GMFCS B	6 (85.7%)	1.5	1 (14.3%)	0.2	0 (0.0%)	−1.1	0 (0.0%)	−1.2	7 (100.0%)
GMFCS C	26 (68.4%)	1.6	6 (15.8%)	1	1 (2.6%)	−2.5	5 (13.2%)	−0.7	38 (100.0%)
In total	60 (58.3%)		12 (11.7%)		14 (13.6%)		17 (16.5%)		103 (100.0%)
**F. The Abnormal Size of the Head—Dysmorphological Classification (hc and HCI) and Level of GMFCS I–V in Subgroup with CP**									
***p* = 0.162** **Cp = 0.54**	**Relative Microcephaly** ***N* (*N*%)** **ASR**		**Absolute Microcephaly** ***N* (*N*%)** **ASR**		**Relative Macrocephaly** ***N* (*N*%)** **ASR**		**Absolute Macrocephaly** ***N* (*N*%)** **ASR**		**In Total**
GMFCS I	3 (50.0%)	−1.6	0 (0.0%)	−1	1 (16.7%)	1.5	2 (33.3%)	2.6	6 (100.0%)
GMFCS II	8 (72.7%)	−0.3	2 (18.2%)	0.7	1 (9.1%)	0.8	0 (0.0%)	−1.1	11 (100.0%)
GMFCS III	5 (100.0%)	1.4	0 (0.0%)	−0.9	0 (0.0%)	−0.5	0 (0.0%)	−0.7	5 (100.0%)
GMFCS IV	3 (75.0%)	0	0 (0.0%)	−0.8	0 (0.0%)	−0.5	1 (25.0%)	1.4	4 (100.0%)
GMFCS V	12 (80.0%)	0.5	3 (20.0%)	1.2	0 (0.0%)	−1.1	0 (0.0%)	−1.4	15 (100.0%)
In total	31 (75.6%)		5 (12.2%)		2 (4.9%)		3 (7.3%)		41 (100.0%)
**G. The Abnormal Size of the Head—Traditional Classification (hc and HCI) and Level of GMFCS I–V in Subgroup with CP**									
***p* = 0.124** **Cp = 0.44**	**Relative Microcephaly** ***N* (*N*%)** **ASR**		**Absolute Microcephaly** ***N* (*N*%)** **ASR**		**Relative Macrocephaly** ***N* (*N*%)** **ASR**		**Absolute Macrocephaly** ***N* (*N*%)** **ASR**		**In Total**
GMFCS I	6 (46.2%)	−1.3	1 (7.7%)	−1	2 (15.4%)	0.5	4 (30.8%)	2.8	13 (100.0%)
GMFCS II	12 (52.2%)	−1.2	4 (17.4%)	0.1	5 (21.7%)	2	2 (8.7%)	−0.2	23 (100.0%)
GMFCS III	5 (83.3%)	1.1	1 (16.7%)	0	0 (0.0%)	−0.9	0 (0.0%)	−0.8	6 (100.0%)
GMFCS IV	5 (62.5%)	0	1 (12.5%)	−0.3	1 (12.5%)	0.1	1 (12.5%)	0.3	8 (100.0%)
GMFCS V	17 (77.3%)	1.7	5 (22.7%)	0.9	0 (0.0%)	−2	0 (0.0%)	−1.8	22 (100.0%)
In total	45 (62.5%)		12 (16.7%)		8 (11.1%)		7 (9.7%)		72 (100.0%)
**H. The Abnormal Size of the Head—Dysmorphological Classification (hc and HCI) and Level of GMFCS A–C in Subgroup with CP**									
***p* = 0.511** **Cp = 0.34**	**Relative Microcephaly** ***N* (*N*%)** **ASR**		**Absolute Microcephaly** ***N* (*N*%)** **ASR**		**Relative Macrocephaly** ***N* (*N*%)** **ASR**		**Absolute Macrocephaly** ***N* (*N*%)** **ASR**		**In Total**
GMFCS A	11 (64.7%)	−1.4	2 (11.8%)	−0.1	2 (11.8%)	1.7	2 (11.8%)	0.9	17 (100.0%)
GMFCS B	5 (100.0%)	1.4	0 (0.0%)	−0.9	0 (0.0%)	−0.5	0 (0.0%)	−0.7	5 (100.0%)
GMFCS C	15 (78.9%)	0.5	3 (15.8%)	0.7	0 (0.0%)	−1.3	1 (5.3%)	−0.5	19 (100.0%)
In total	31 (75.6%)		5 (12.2%)		2 (4.9%)		3 (7.3%)		41 (100.0%)
**I. The abnormal size of the Head—Traditional Classification (hc and HCI) and Level of GMFCS A-C in Subgroup with CP**									
***p* = 0.108** **Cp = 0.36**	**Relative Microcephaly** ***N* (*N*%)** **ASR**		**Absolute Microcephaly** ***N* (*N*%)** **ASR**		**Relative Macrocephaly** ***N* (*N*%)** **ASR**		**Absolute Macrocephaly** ***N* (*N*%)** **ASR**		**In Total**
GMFCS A	18 (50.0%)	−2.2	5 (13.9%)	−0.6	7 (19.4%)	2.2	6 (16.7%)	2	36 (100.0%)
GMFCS B	5 (83.3%)	1.1	1 (16.7%)	0	0 (0.0%)	−0.9	0 (0.0%)	−0.8	6 (100.0%)
GMFCS C	22 (73.3%)	1.6	6 (20.0%)	0.6	1 (3.3%)	−1.8	1 (3.3%)	−1.5	30 (100.0%)
In total	45 (62.5%)		12 (16.7%)		8 (11.1%)		7 (9.7%)		72 (100.0%)

*N*—numbers of patients, %—percent, *p*—probability value calculated by chi-square test of independence, Cp—Pearson’s Contingency Coefficient C, Cp ≥ 0, when is distant from 0, there is some relationship and the closer to 1, a perfect association will approach, ASR—Adjusted Standardized Residuals, values >1.96 reflect a greater number, and those below <−1.96 correspond to a smaller number than random distribution.

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
