# Peer review of "Abnormal Head Size in Children and Adolescents with Congenital Nervous System Disorders or Neurological Syndromes with One or More Neurodysfunction Visible since Infancy"

_jcm, 2020, doi:10.3390/jcm9113739_

Round 1

Reviewer 1 Report

The authors have addressed all the issues I reported in the first review.

Reviewer 2 Report

The authors tried to address the concerns stated by the reviewers; however, I still feel that the article is ill-focused and am unsure what the novelty is.

This manuscript is a resubmission of an earlier submission. The following is a list of the peer review reports and author responses from that submission.

Round 1

Reviewer 1 Report

Dear editor,

thank you for inviting me to review the manuscript submitted by Perenc et al. The authors report head circumference (HC) data and further anthropometric data in a retrospective study based on a cohort of 327 children and adolescents / young adults, ages 4-18. The aim to identify a relationship between the HC and further disease aspects.

Though the idea behind the study is good, the design is - in my view - not adequate to state the conclusions that the authors write in their paper. The cohort is very heterogenous, with only individual patients in some subgroups. I suggest that the authors rewrite their study focussing only on the largest group - the CP group which in itself is already heterogenous. They could report the presence of microcephaly and macrocephaly (relative and absolute), IQ values, height, weight, GMFCS, schooling and correlate these values. Why is it interesting to correlate dysmophology criteria with head size? Isn’t it more of interest to generate data on a (pseudo-)specific subgroup of neurologic disease/disorder in childhood?

The paper also needs in-depth critical word-for-word rethinking and rewriting. Already in the introduction the authors state that ‘microcephaly and macrocephaly are developmental disorders of the skull….’ but I would argue that these are clinical findings rather than diseases and would also rethink calling them ‘disease of the skull’ because the origin is usually not a ‘bone problem’. Also in the second sentence, they state that the HC estimates the size of the skull. I would rather indicate that the HC is an indirect measure of the brain / head size. Also, why concentrate on the definitions in the introduction rather than on the main point that the authors would like to analyze the distribution of HC in a specific population. I am missing a figure showing the distribution. These sort of thoughts go through my mind throughout the paper.   I am happy to review a revised version.   With kind regards,

Reviewer 2 Report

Overall the manuscript is not clear. There is too much information/results and it is hard to follow the logic of the authors.

From a methodological point of view, as it is a retrospective study, the measurement of the head circumference for each child was probably not realized by the same physician. Or it is known that the HC can vary a lot between two different measures done by two different physicians/nurses. Therefore, this is a very strong bias. Also, the rationale for using z-score for HC is not clear.

The overall conclusion is not new. It is known that children with neurological conditions often have a "small head". With such a large cohort of patient, I am sure the authors could have produced something more useful to the scientific community.